# Understanding the Relationship between Water Availability and Biosilica Accumulation in Selected *C*_4_ Crop Leaves: An Experimental Approach

**DOI:** 10.3390/plants11081019

**Published:** 2022-04-08

**Authors:** Francesca D’Agostini, Vincent Vadez, Jana Kholova, Javier Ruiz-Pérez, Marco Madella, Carla Lancelotti

**Affiliations:** 1CaSEs Research Group, Department of Humanities, University Pompeu Fabra, c/Ramon Trias Fargas 25-27, 08005 Barcelona, Spain; javier.ruizperez.academic@gmail.com (J.R.-P.); marco.madella@upf.edu (M.M.); carla.lancelotti@upf.edu (C.L.); 2DIADE Unit, IRD, University of Montpellier, Av. Agropolis 911, 34394 Montpellier, France; vincent.vadez@ird.fr; 3Crop Physiology Laboratory, ICRISAT, Patancheru 502324, India; j.kholova@cgiar.org; 4Department of Information Technologies, Faculty of Economics and Management, Czech University of Life Sciences, Kamýcká 129, 165 00 Prague, Czech Republic; 5ICREA-Catalan Institution for Research and Advanced Studies, Pg. Lluís Companys 23, 08010 Barcelona, Spain; 6Department of Geography, Archaeology and Environmental Studies, University of Witwatersrand, 1 Jan Smuts Avenue, Braamfontein, Johannesburg 2000, South Africa

**Keywords:** biosilica, phyoliths, water availability, *C*_4_ crops

## Abstract

Biosilica accumulation in plant tissues is related to the transpiration stream, which in turn depends on water availability. Nevertheless, the debate on whether genetically and environmentally controlled mechanisms of biosilica deposition are directly connected to water availability is still open. We aim at clarifying the system which leads to the deposition of biosilica in *Sorghum bicolor*, *Pennisetum glaucum*, and *Eleusine coracana*, expanding our understanding of the physiological role of silicon in crops well-adapted to arid environments, and simultaneously advancing the research in archaeological and paleoenvironmental studies. We cultivated ten traditional landraces for each crop in lysimeters, simulating irrigated and rain-fed scenarios in arid contexts. The percentage of biosilica accumulated in leaves indicates that both well-watered millet species deposited more biosilica than the water-stressed ones. By contrast, sorghum accumulated more biosilica with respect to the other two species, and biosilica accumulation was independent of the water regime. The water treatment alone did not explain either the variability of the assemblage or the differences in the biosilica accumulation. Hence, we hypothesize that genetics influence the variability substantially. These results demonstrate that biosilica accumulation differs among and within $C_{4}$ species and that water availability is not the only driver in this process.

## 1. Introduction

Silicon uptake, in the form of monosilicic acid $Si{\left(OH\right)}_{4}$, depends to a great extent on water availability in the soil [1]. Silicon is radially taken up by root cortical cells either by diffusion (passive) or in an energy-dependent (active) manner [2]. A channel-type Si transporter gene, Lsi1, translocates Si across the plasma membrane from apoplast to cells [3], and Si is transported to proximal side apoplastic connections by the efflux transporter Lsi2 [4]. The xylem loading is mediated by another silicon transporter gene, Lsi6, found in the xylem parenchyma cells of the leaf sheath and blades [5]. After roots uptake $Si{\left(OH\right)}_{4}$, it is transported up to the shoots via the xylem, with the transpiration stream acting as the main motive force [6]. This model has been tested in rice (*Oryza sativa* L.), but the same mechanism has been discovered to be dependent on the three transporter genes located in different cell layers in maize (*Zea Mays* L.) [7], and in several vegetables such as pumpkin [8] and cucumber [9]. Far less is known of this mechanism in $C_{4}$ species. Once in the plant tissues, monosilicic acid precipitates, forming phytoliths in the cell wall or in the cell lumen [10]. Phytoliths are solid deposits of amorphous biosilica ($SiO_{2}$·$nH_{2}O$) produced by living plants in and among cells [11]. In grasses, three principal mechanisms of silicification have been described: (a) a passive cell wall silicification driven by dehydration; (b) a spontaneous silica deposition driven by transpiration in the cell lumen, after the cell protoplast death, and (c) a controlled mechanism that can occur both in the cell wall or in the external side of the functional plasma membrane, templated by cell wall polymers or materials which enhance silica deposition. This mechanism characterizes the silicification of the so-called silica cells, epidermal cells filled up with silica independently of transpiration [12]. In tissues with photosynthetic activity, like leaves and to a minor degree culms, silicon polymerization due to supersaturation by transpiration-driven water loss (options a and b) seems to play a fundamental role in silica accumulation [13]. However, although transpiration has been investigated as one of the main factors, producing high concentrations of silicic acid in plant shoots [14], it is clear that it is not the only parameter that accounts for biosilica precipitation, and that localized silica deposition forming “silica cells” involves other factors (mainly compounds which enhance silica deposition) [12,15,16]. The role and function of Si in plants is not yet fully understood, and this impairs our comprehension of the mechanisms that drive its deposition. However, since the trait persists, it should confer some advantage in improved fitness [17].

Si accumulation in plant tissues varies from 0.1% up to 10% of the dry weight [18]. Even though the precise mechanical properties of Si remain elusive, it has recently been argued that Si accumulation has little if any intracellular role [19], while the biosilicification of phytoliths can act as a protection against numerous environmental stresses [6]. It has been suggested that Si could play a fundamental role in response to water stress conditions: changing the hydraulic conductivity and the osmotic adjustment [20]; forming a thin layer in the epidermal cells which gives rigidity to the tissues [21]; protecting the veins to keep water supply running in limited water conditions [22]; providing better light inception leading towards a better assimilation rate and chlorophyll biosynthesis [23,24]; reducing epidermal water losses through the formation of a Si-cuticule double layer [25]; and increasing water use efficiency in response to the impact on the stomatal movement [26]. Silicon also has a positive effect on mineral nutrient balance, improving resistance to heavy metal stresses and water storage by diluting salts in the cells [27,28]. Furthermore, silicon has been proven to have a strong impact in preventing pathogen infection [29,30].

Recently, phytoliths have been suggested as proxies to determine plant water availability during plant growth since the ratio between environmental morphotypes (transpiration-driven options a and b) and genetically controlled ones (produced in every water context and enhanced by activator substances-option c) is supposed to change under different water regimes [31,32]. We argue that before investigating phytolith morphotype assemblage compositions, which are connected to an intrinsic cell-type silicification mechanism, it is essential to clarify what the dynamics are that determine the accumulation of biosilica in grasses and if transpiration is a major driver in Si uptake and distribution. Our objective was to investigate whether or not there was a strong species-specific or genotypic variation in silica accumulation and if transpiration could account for most of the biosilica produced by the crops.

$C_{4}$ species are relatively few compared with the $C_{3}$ plants, but they account for approximately 25% of the primary production on the planet since they dominate savannah and grassland biomes of warm-temperate to tropical and arid regions [33]. Their ability to withstand high temperatures and scarce and erratic rainfall patterns is strictly connected to their photosynthetic pathway. $C_{4}$ photosynthesis increases the assimilation rate by concentrating $CO_{2}$ at the site of rubisco using a structural mechanism that is distributed between two compartments: the mesophyll and the bundle sheath cells (Kranz anatomy) [34]. This characteristic morphology reduces photorespiration and allows rubisco to operate close to its optimum even with a reduced stomatal opening, which prevents water loss but reduces $CO_{2}$ concentration. This mechanism has an energetic cost (higher request of ATP to fix $CO_{2}$ into organic acids) that $C_{4}$ crops can afford since they often develop in high-radiation environments [35], which allows them to compensate with a higher ATP supply. Indeed, $C_{4}$ plants are specifically adapted to xeric environments [36]. Due to the complexity of $C_{4}$ physiological reactions to water saving, there is still no consensus on whether water availability during plant growth can be inferred through the study of their organs, such as the grains (e.g., for the discrimination of ${}^{13}$C) or through phytoliths (e.g., $\delta^{29}$Si discrimination or morphotype ratios) [37,38]. Conversely, these same methodologies have already been validated for $C_{3}$ species, which are characterized by less complex water and $CO_{2}$ cycles [31,39]. Indeed, the beneficial effects of Si in different plants is well-documented on $C_{3}$ species such as *Triticum durum* (Desf.) [22] or *Triticum aestivum* (L.) [27]. More work remains to be done for $C_{4}$ crops. The few studies conducted on sorghum [40] and maize [41] demonstrate that silicon could enhance drought tolerance even in crops well-adapted to survive stress conditions. Thus, understanding the role of Si on $C_{4}$ crops’ growth and productivity is important.

For this study, we chose *Sorghum bicolor* (L.) Moench (sorghum), *Eleusine coracana* (L.) Gaertn (finger millet) and *Pennisetum glaucum* (L.) R.Br. (pearl millet). Sorghum and millet varieties have been ranked by the FAO among the 150 top crops produced globally [42]. Sorghum and millets are known to be well-adapted to water-scarce situations, having a relatively short crop cycle and requiring modest water amounts [43], but they differ in their $C_{4}$ sub-pathway: sorghum is a NADP-ME species, whereas finger and pearl millets are NAD-ME subtypes [44]. NAD-ME and NADP-ME species differ in their typical Kranz anatomy, in their metabolite flow through mesophyll and bundle sheath cells, and in their plastid transporters [45]. Since the biochemical differences between the two pathways involve both the water use adaptation and photosynthetic abilities, testing both pathways allowed us to assess all the possible intra- and inter-pathway differences in biosilica accumulation.

In addition, while we can rely on data that associate sorghum with a transporter-driven Si accumulator, we have no information on the role of genetics for Si uptake in *Pennisetum glaucum* and *Eleusine coracana*. The role of silica in millets remains, therefore, to be clarified, and it is interesting to test whether it relies on genetics or on a transpiration-dependent mechanism. Indeed, *Sorghum bicolor* has been tested for the Lsi1 protein sequence by Vatansever and colleagues [46], who discovered two homologous genes codifying for the transporter proteins. A recent phylogenetic analysis of silicon transporters across the biological kingdoms shows a high level of conservation of Lsi2 in embryophytes [19], indicating an early evolution, which lets us suppose its presence in $C_{4}$ species. The study identified the presence of 5 Lsi2 homologous in *Sorghum bicolor* but highlighted the complexity and the diversity of Lsi2 transporters, leaving open different hypotheses for millets. In sorghum, Slp1 has also been localized; it is a unique amino acid compositional protein involved in the precipitation of silica in the silica cells [47]. These data suggest that Si plays a fundamental role in sorghum physiology since its deposition seems to be strictly regulated.

Reports on controlled experimental cultivations to test silica content in relation with water availability are more abundant for $C_{3}$ species than for $C_{4}$; indeed, rice and vegetables are the most cited in literature [48]. Models and existing data suggest that biosilica formation could be more sensitive to water availability in $C_{3}$ species since $C_{4}$ grasses are characterized by a specific physiological mechanism that prevents water loss and reduces the transpiration rate, which has been shown to force silica uptake and distribution along the plant [49]. Previous studies have found positive correlations between water availability and silica content in the form of phytoliths, both in $C_{3}$ and in $C_{4}$ species [14,50,51]. These studies were performed on crops grown under controlled environmental conditions indoors [50], in a greenhouse [14] or in outdoor fields, where it is difficult to keep water availability and transpiration rate under control [51]. Only Jenkins and colleagues [31,51] included more than one genotype of the same species in the experimental replication. Laboratory experiments tend to demonstrate that plant Si content is higher with high water availability but in natural ecosystems many co-variables, including herbivore pressure, higher light intensity than in glasshouse conditions, different atmospheric conditions (usually higher evaporative demand), and nutrient availability, can influence silica content [17]. To mitigate the loss of the co-variable effect in pot/greenhouse experiments but still be able to evaluate weekly plant transpiration, we set our experimental cultivation outdoors in lysimeters with ten different genotypes tested for each species.

In this paper, we aim to clarify the factors involved in silica accumulation and their variations in selected $C_{4}$ species and, in addition, provide an important reference for archaeological and paleoenvironmental phytolith studies and agronomic research. To achieve our goal we decided to test different landraces of pearl millet, finger millet and sorghum grown in different water conditions for biosilica accumulation. Our objective was to highlight any difference among the three $C_{4}$ species and between two different water regimes which correspond to an extreme water stress and an optimum irrigation condition.

## 2. Results

### 2.1. Plant Growth

All replications grew and produced leaves. Finger millet plants did not produce any panicles either in well-watered (WW hereafter) or in water stressed (WS hereafter) conditions (data available in Appendix A). However, WW replications of finger millet produced an abundant canopy (see Table 1), possibly indicating that biomass allocation to reproductive structures was replaced by vigorous vegetative growth. Indeed, finger millet plants transpired more and produced more biomass in WW conditions with respect to the other two species (Figure 1). Biomass has been calculated as the sum of the weights of all the plant components: stem, leaves, panicles and tillers (data and boxplots available in Appendix A). Sorghum produced more leaves and more biomass on average in WS conditions (Table 1) than the other two species, and it had higher values of transpiration efficiency (biomass produced (g)/water transpired (L)) (Figure 1 and Figure 2 and Table 1). Sorghum plants grew for an average of 82 days, pearl millet for 73 and finger millet for 105 days. Sorghum plants flowered on average in 31 ± 20 days in WS and 39 ± 18 days in WW (5–6 weeks). Pearl millet flowered on average in 33 ± 21 days in WS and 46 ± 14 days in WW (6–7 weeks). Finger millet did not flower at all.

### 2.2. Silica in Leaves

Sorghum was by far the species that accumulated the most biosilica, especially under WS conditions (between 5.07% and 11.88% of the dry weight), 2 times more with respect to pearl millet (2.67–6.87%) and 4 times more than finger millet (1.29–3.01%). Moreover, sorghum accumulated almost the same amount of biosilica in WS (mean and SD = 7.94 ± 1.96%) and WW (mean and SD = 8.42 ± 2.93%) conditions, while the other two species of millet presented almost double the percentage in WW conditions (pearl millet WW mean and SD = 7.51 ± 2.35%, WS mean and SD = 4.6 ± 1.47%; finger millet WW mean and SD = 3.74 ± 1.27%, WS mean and SD = 2.37 ± 0.55%) (Figure 3). Considering all the landraces, finger millet and pearl millet accumulated more biosilica in WW conditions with respect to the WS treatment, and the difference is statistically significant (adjusted *p*-value pearl millet WW-pearl millet WS = 0.0006 and finger millet WW-finger millet WS = $1.78\times 10^{-5}$ ). By contrast, the difference in biosilica accumulation between the two treatments in sorghum is not significant, with an adjusted *p*-value of 0.6021.

The effect of water treatment on biosilica accumulation in each genotype is shown in Figure 4. Most of the finger millet and pearl millet genotypes (i.e., FM2, FM6, FM7, PM5, PM7, PM9) had statistically significantly higher biosilica content in the WW treatment with respect to WS. For the remaining landraces of millets, the treatment did not have a significant effect on the biosilica accumulation, although they followed the same trend, accumulating more in well-watered conditions. By contrast, sorghum landraces had almost the same value of biosilica accumulation in WW and WS conditions apart from S6, which accumulated less in WW conditions, and S8, which accumulated less in WS conditions. Finger millet and pearl millet displayed low variance of biosilica accumulation across genotypes both in WW and in WS condition ($\sigma^{2}$ of finger millet WW = 1.33, finger millet WS = 0.30; pearl millet WW = 3.90, pearl millet WS = 3.90). By contrast, in sorghum, there is a lot of variation among genotypes, especially in WW conditions ($\sigma^{2}$ of sorghum WW = 8.92, sorghum WS = 3.03).

We found a significant relationship between silica accumulated by finger millet and pearl millet accessions and the total water transpired by the plants (Figure 5), while the relationship was not significant for sorghum. In finger millet and pearl millet, biosilica accumulation followed the same trend even if we analyze the different landraces separately (regressions available in the Appendix A), showing a comparable response to water treatments for all of them. On the contrary, sorghum landraces showed a more variable response to total water transpired: positive for S8 (from Pakistan) and negative for S6 (from Sudan). The three remaining landraces did not show significant interaction with the total water transpired.

Since the relationship between total water transpired and total silica extracted (%) was significant but not high, we decided to compare different predictors to observe which had the strongest effect on biosilica accumulation. The model that explained the highest variability in the dataset included the interaction effect of water treatment and genotype (Table 2). When the dataset with all the accessions was tested for the effect of water treatment alone, the explained variability was too low (5%) to accept the model, showing that biosilica accumulation was not exclusively associated with the water regime. The explained variability increased substantially when the species was used as an explanatory variable. The value of the adjusted R-squared increased by 10% when the genotypes were used as predictors and by 26% when the interactive effect of genotypes and treatment was used, showing that there was a high variability among genotypes and in how genotypes responded to water treatment.

### 2.3. Silica Extracted and Physiological Parameters

We used a redundancy analysis to test which physiological parameters, including biosilica accumulation, characterize the response of the plants to the water treatments (Figure 6). The triplot captures ca. 50% of the variance and shows a gradient where WW samples (on the left side of the triplot) are characterized by higher values of biomass, leaf biomass (“Leaves”), stem biomass (“Stem”), biosilica accumulation (“SilicaPercentage”), panicle biomass (“Panicle”) and flowering time (“Flowering”), while WS samples (on the right side of the triplot) produced more leaves in comparison to the total biomass (“LeavesPercentage”) and have higher values of transpiration efficiency.

## 3. Discussion

The results from the phytolith extraction suggest that sorghum accumulates more biosilica with respect to the two species of millet. Pearl millet and finger millet show a positive relationship between total water transpired and biosilica accumulated. On the contrary, overall sorghum production of biosilica is not influenced by the water treatment. The water treatment alone does not explain either the physiological response of the plants or the differences in biosilica accumulation. The genetics related to the different genotypes influence the variability substantially.

### 3.1. Biosilica Accumulation, Transpiration and Water Availability

Biosilica accumulation was positively related to total water transpired in pearl millet and finger millet for all the accessions under study (Figure 3, Figure 4 and Figure 5). This difference could be explained by a passive (transpiration-driven) silica accumulation mechanism, controlled by the transpiration stream in the shoots of millets. We hypothesize that millets probably lack an energy-dependent transport process which loads the silica in the xylem, facilitating the influx and efflux out of the cortical cells [4] even when the transpiration rate is low. Thus, the accumulation of Si in the leaves may be mostly related to environmental parameters such as the water transpired, which involves the passive diffusions of the silicic acid with water and other uncharged solutes [4]. This finding implies that the silica accumulated by millets could be a good proxy for plant water availability. By contrast, sorghum landraces showed a null correlation with total water transpired, and no differences in silica accumulation between well-watered and water-stressed replications can be observed (Figure 3, Figure 4 and Figure 5). This result suggests the presence, in sorghum, of an active mechanism that transports the silicic acid actively up to the leaves independently of the water conditions (transporter-governed Si uptake). We may speculate that this amount of silica is physiologically useful for the plant and could play a role in giving structural support to the tissues [21,22,25], contributing to the osmotic adjustment and the biosynthetic mechanisms [20,23,24,26], and/or influencing the mineral balance by protecting tissues from toxic elements and insect/fungi attack [27,28,29,30]. As a consequence, the transfer, mediated by the presence of the transporter Lsi1 and Lsi2 (and Lsi6) is maintained efficiently and constantly [15,46]. These findings support previous results by Verma and colleagues [52] that showed how higher levels of Si accumulation are associated with better performance in water deficit conditions, as it increases photosynthetic leaf gas exchange and improves plant biomass [52]. In conclusion, while silica accumulation, and consequently phytolith production (i.e., number of silicified cells), in sorghum is not a good proxy for water availability, it may have a physiological role that explains its relatively limited biomass reduction under WS conditions.

### 3.2. Biosilica Accumulation and Genotypic Variation

The multiple linear regression models showed that water availability was not the only factor that accounted for the variability in biosilica accumulation (Table 2). Species and genotypes predicted the amount of silica accumulated better than the treatment alone, as it was previously suggested that phytolith production was influenced more by the phylogenetic position rather than by environmental effects [53]. However, it is interesting to note that our results show that biosilica accumulation is best explained by the interactive effect of genotypes and watering, suggesting that both variables had a substantial effect in determining the variability in biosilica accumulation. Si uptake and distribution in grasses is a dynamic process that depends on Si availability [54], the plant’s internal Si demand (physiology) [54], the expression gene involved in Si uptake and accumulation along the plant canopy [55], and external factors which affect the silica distribution, such as the transpiration stream that drives the passive uptake [56]. In the previous section, we argued that the absence of an active mechanism of Si uptake, controlled by genetics, could make the environmental effect prevail in finger and pearl millet. The fact that all landraces of millets analyzed in this work display comparable responses (Figure 3 and Figure 5) supports the hypothesis that transpiration is the variable that played the most important role in biosilica accumulation. The presence of Si transporters in sorghum has been previously suggested [57], and our results show that silica accumulation exceeded the transpiration stream rather than being dependent on it for most of the landraces (Figure 3 and Figure 5). Indeed, we prove the existence of a substantial difference in biosilica accumulation between millets and sorghum that clarifies why the effect of species and genotypes determined the silica accumulation better than the treatment alone. The strong effect that genotypes have in explaining the variability in the dataset (Table 2) could also be related to the specific genetic mechanism that sorghum landraces display for biosilica accumulation. The wide expression profile of Lsi1 and Lsi2 transporter proteins and their regulation, which can be related to different absorption rates of monosilicic acid, is known in the literature [58]. Furthermore, the presence of mutants unable to absorb high quantities of silica in sorghum has been suggested by Markovitch and colleagues [59], which possibly justifies the low biosilica accumulation of the landrace S6 (Figure 4). While silica accumulation in millets responds to the water availability, in sorghum, there is an extra variable that could play a role in biosilica formation and that could account for the high variability rate: the genetics of the silicon transporter.

### 3.3. Biosilica Accumulation and Plant Physiology

The redundancy analysis triplot (Figure 6) showed two important trends: WW samples accumulate more biomass, and WS replications are more efficient in terms of transpiration (“TranspirationEfficiency”). Biosilica accumulation is positively related to biomass accumulation, and it increased when the plant was heavier, affecting the relation of biosilica with transpiration efficiency. Therefore we hypothesize that the effect of the biomass on the biosilica accumulation is consistent, and we argue that it should be related to the effect that the treatment has on the biomass accumulation. Indeed, the RDA triplot revealed that the overall water transpired by the plants has a direct effect on the physiological response (combination of the physiological parameters, including biosilica accumulation) of the landraces under study and, as a consequence, that biosilica accumulation is a parameter that could be taken into consideration to distinguish the two water treatments. As we argued above, Si accumulation in the leaves could possibly play a role (a) to retain leaf water potential by influencing the stomata functioning that silicify more in WW conditions [49], or (b) to defend the leaf surface from biotic and abiotic stress agents [60,61,62,63,64], contributing actively to the better performance of the WW replications. These results seem to indicate that the biosilica accumulation process is stimulated by the physiological conditions and the developmental stage of the plants, as suggested by Mitani and Ma [65]. We need to highlight that this trend has exceptions represented by most of the landraces of sorghum, which accumulate more biomass in WW conditions, but their biosilica accumulation pattern did not change between the two treatments (explanation above). Nevertheless, the fact that most sorghum landraces showed a constant rate of Si absorption independent from the water treatment does not exclude the hypothesis that Si plays a role on the physiological performance of the plant; on the contrary, it validates this hypothesis once more,.

## 4. Materials and Methods

### 4.1. Selection of Landraces

To exclude the possibility of changes in the genetics of Si absorption introduced in the breeding process, we decided to work with traditional landraces to observe the variability in biosilica production in un-improved genotypes. Landraces were chosen according to two criteria: (a) area of origin, under the assumption that we need to investigate dryland crops which might have suffered different local adaptations and evolution processes, and (b) climate of origin, to cover all the possible variability within an arid environment. Thus, samples were chosen by selecting their coordinates of origin using the Climatic Research Unit TS3.10 Dataset [66] and https://en.climate-data.org/ (accessed on 1 October 2018), which provide high resolution data on world temperature, precipitation and relative humidity. The databases provide five independent climate variables (mean temperature, average sun hours, precipitation, rainy days and relative humidity) covering the global land surface. They also estimate maximum and minimum temperatures as well as secondary variables, such as potential evapotranspiration. CRU TS3.10 has been used to intersect the climate variables (provided in the form of grids) with the origin coordinates of the varieties on Google Earth. We obtained landraces of pearl millet (*Pennisetum glaucum* L.R.Br), finger millet (*Eleusine coracana* L.Gaertn) and sorghum (*Sorghum bicolor* L.Moench) seeds from the collection of the ICRISAT (Hyderabad, India) gene bank (Table 3).

### 4.2. Experimental Cultivation in Lysimeters

Plants were cultivated in lysimeters (PVC tubes of 2 m in height and 30 cm in diameter) inside concrete pits in ICRISAT, Patancheru, India (17°31${}^{\prime }$ N 78°16${}^{\prime }$ E). The lysimeters were filled with a mixture of 1:1 Alfisol-Vertisol, and several crops had been cultivated previously in the same soil mixture inside the lysimeters. At the time of the experiment, the lysimeters filled with the same soil had been fallowed for about 8 months. As the lysimeters were placed outdoors (but covered by a rain-out shelter in case of rain), the experimental conditions allowed for simulating real field conditions regarding plant spacing (11 plants m${}^{-2}$ ), soil availability for water exploration (2 m depth of soil available for each plant) and general growing conditions. In addition, lysimeters provide control over water availability and transpiration rates thanks to their regular weighing, which allows transpiration assessment and possible re-watering to the desired soil water content levels [67]. The facility was chosen to produce results that can be compared with plants grown in a real field, which make sense both from an archaeological and an agriculture point of view, while the water availability and the transpiration rate have been kept under control to answer the experimental question. While the plants were in the early stages of development, cylinders were watered regularly to keep them close to 100% field capacity. When the plants had grown to c. 20 cm, about three weeks after planting, they were watered to field capacity, and then the soil surface was covered with a plastic sheet and 2 cm of low-density polyethylene granules, in order to prevent about 90% of the soil evaporation [67]. After this, the lysimeters were weighed weekly to evaluate plant water loss from transpiration [68]. Experimental cultivation in pots offer the possibility to collect reliable data only up to an early vegetative growth stage, beyond which the soil volume of the pots, e.g., 1 L or 2 L, becomes limiting to root growth [69] and the pots are no longer sufficient to ensure a normal development of the plant. Several reports and research articles proposed results based on the use of pots for experimental cultivation [70,71], and although we recognize the value of these studies, we propose an alternative methodology that can be exploited to achieve trustworthy results in crop physiology investigation as in experimental archaeology.

The experimental design included ten different landraces for each of the three species to obtain a sufficient sample size to observe the physiological parameters of the growing crops. Five genotypes of sorghum, five of pearl millet and four of finger millet were selected for the present study (Table 3). Selection was based on the physiological response to watering. The genotypes with the highest diversity in physiological parameters were selected within each species in order to assess inter-genotype variations in biosilica accumulation. The remaining 16 landraces have been cropped and stored for future analysis. To represent the range of possible water scenarios, two different water managements were tested to simulate water status in (a) rain-fed conditions in arid environments (water-stressed) and in (b) an irrigated system that acted as control (well-watered). We set five replications for each landrace and treatment, where one replication consisted of one lysimeter in which two plants were grown. The experiment followed a complete randomized block design with species-water treatment as the main block and genotypes as the sub-factor randomized within each block. Genotype replicates were randomized in the pits in order to prevent unintended environmental effects (e.g., heat gradient from the pit walls). Cropping took place in the period between February and May 2019. The complete experimental design is available in the Appendix A. During growth, lysimeters were weighed weekly, allowing the measurement of the transpiration rate and the application of water according to treatment. Flowering dates were also recorded, and biomass was measured after harvest. The physiological parameters measured and available in the Appendix A are: total water used (TWU), which corresponds to the total water transpired by the plant in liters; total water added (TWA), corresponding to the liters given to the plant during irrigation; “Biomass”, which is the sum of stem, leaves, panicles and seeds weights for each plant in grams; “Leaves”, which is their weight (in grams); “LeavesPercentage”, which consists of the percentage of leaves with respect to the total weight of the plant; “Stem” weight (in grams); “Panicle” weight, corresponding to the chaff and seed weight in grams; “Flowering” time, representing the number of days that each plant took to bloom; and “Transpiration Efficiency”, which is the rate between biomass in grams/water transpired in liters. Temperature and relative humidity were collected every 30 min by 2 recorders (Gemini Tinytag Ultra 2 TGU-4500 Datalogger) placed in the two different pits in the crop canopy. Temperature was maintained at 32.28 ± 0.10 ${}^{\circ }$C, and relative humidity was maintained at 42.57 ± 0.23% Rh (measurements are available in Appendix A).

WW plants were irrigated weekly till maturity to maintain the crop at 80% of the soil field capacity, which is optimal for plant growth. WS replications received water until the flowering time to simulate the rain-fed scenario, when water is available only at the beginning of the life cycle but is scarce during the reproductive stage [72]. WW plants have been watered once per week: sorghum plants received an average of 34.99 ± 0.93 L in total, pearl millet received 35.80 ± 1.16 L, and finger millet received 48.93 ± 0.82 L. WS plants received 11 L each across species, distributed in the first 2 months after sowing and before stress imposition. Taking into consideration the diameter of the cylinders (30 cm), it roughly matches to 153 mm of water. Precise data about total water added are available in the Appendix A. An ethnographic investigation by Lancelotti and colleagues [38] recorded instances where pearl millet was grown in a terminal water stress condition (when water scarcity affects the reproduction stage and the grain filling), which is an extremely common growth condition in semi-arid tropics. To simulate this field condition, we set a late-water-stress imposition experiment [73]. It is known that the reproductive stage is particularly sensitive to water deficits and that water availability during and after anthesis is critical [74], so through the imposition of a late water stress, we simulated a realistic rain-fed scenario. WW plants were harvested when the panicles were mature, and the WS replications were harvested when their transpiration rate dropped below 10% of the initial value, indicating full stomatal closure (and further water losses only due to cuticular conductance) [75,76]. Harvesting was done according to genotype, i.e., we harvested all the plants of the same genotype on the same day, when at least $\frac{3}{5}$ replications reached maturity.

### 4.3. Extraction Method from Leaf Samples

Practically all silica in plants is found as opal silica. As a consequence, we consider phytoliths good proxies for biosilica accumulation, and we proceeded with their extraction. For each landrace, we selected and analyzed two plants of each of the three replications in both treatments. We selected three replications out of five to reduce the sample size, while counting on more than 30 samples per species, as previously suggested to meet statistical representativeness by Jenkins et al. [31,51]. In total, we analyzed 36 samples of finger millet, 18 WW and 18 WS, and 40 samples of sorghum and pearl millet, with 20 WW and 20 WS each. Leaves are the organs through which the transpiration stream goes and where it is assumed to produce the strongest variability for silica accumulation. Samples were prepared for extraction by washing leaves in an ultrasound bath at room temperature to remove all the external sources of silica, and by letting them dry for two days in a ventilated room. Once completely dried, 0.1 g of leaf material was ashed at 500 ${}^{\circ }$C for 12 h in a ceramic crucible covered with a lid to avoid carbonization and prevent cross-contamination. Carbonates were removed using HCl (10% *v*/*v* ). Organic matter was oxidized with $H_{2}O_{2}$ (10% *v*/*v*) at 40 ${}^{\circ }$C until the reaction subsided. The residue of silica was rinsed with distilled water and dried at 60 ${}^{\circ }$C. When the powder was completely dry and cool, the extract was weighed to measure the biosilica accumulation of each plant. To facilitate direct comparison among samples, biosilica accumulation has been transformed into percentage. The full protocol used for extraction can be found at https://www.protocols.io/view/phytolith-extraction-and-counting-procedure-for-mo-b6streen (uploaded on 1 April 2022).

### 4.4. Statistical Analysis

We evaluated the distribution of silica extracts by performing Shapiro’s test for normality and Lavene’s test for equality of variance [77]. ANOVA, two-way ANOVA and Tukey’s honestly significant difference (HSD) tests have been performed to evaluate the significant difference of the variance of treatments, species and genotypes, to evaluate specific pairwise comparisons between treatments and to compare the models. Linear regressions have been performed to model the relationship between biosilica accumulation and total water transpired. Even though response variables (physiological parameters and silica accumulation) were represented by a bimodal curve (juxtaposition of WW and WS normal distributions), linear regression models were used as the distribution of the model residuals was normal [78]. Multiple linear regression models were used to evaluate the variable/s with the strongest effect on bioslica accumulation. Redundancy analysis (RDA) was applied to summarize the variation of the dataset (including the physiological parameters measured and biosilica accumulation) using water treatment as an explanatory variable and conditioned by species and genotypes (covariables). Since the physiological parameters have different units, and as a consequence they are distributed along specific ranges, they have been scaled to unit variance to avoid variables with high values to have a stronger effect on the analysis [79,80]. Statistical analyses and data visualization were conducted in R (version 3.5.1) using standard functions of base, ggplot2 (version 3.3.5) [81] and vegan (version 2.5.6) [82] packages; all the scripts are available in Appendix A.

## 5. Conclusions

The results presented in this paper allows us to conclude that:Water availability plays a fundamental role in determining biosilica accumulation in finger millet and pearl millet, which seem to be passive accumulators where transpiration-driven biosilica production prevails over genetic-mediated silica deposition. Therefore, we maintain that biosilica accumulation in finger millet and pearl millets is a good proxy for water availability;Based on the results obtained, different sorghum genotypes absorbed and accumulated silica differently. The relatively high magnitude of variability in response to water treatment suggests that biosilica accumulation in sorghum is not a good proxy for plant water availability. Indeed, sorghum is seemingly characterized by a transporter-governed mechanism, which possibly determines a high variability among genotypes. In the literature, the topic is rather controversial. The results of this paper lead to new perspectives, highlighting that not all the sorghum genotypes respond equally to biosilica accumulation;Both environmental conditions and genetic variability play distinct roles in biosilica accumulation, even within the same species.

Nevertheless, we want to highlight that different results have been published, especially in relation to archaeological studies, and this might derive from the experimental settings. Pot-based experiments, possibly conducted under light limitations within glasshouses, may be flawed because of the strong influence of light on transpiration stream and plant development. Therefore, we suggest that experimental cultivation using a standardized methodology is now needed. It is also needed to respond to archaeological and palaeoenvironmental questions.

## Figures and Tables

**Figure 1 plants-11-01019-f001:**
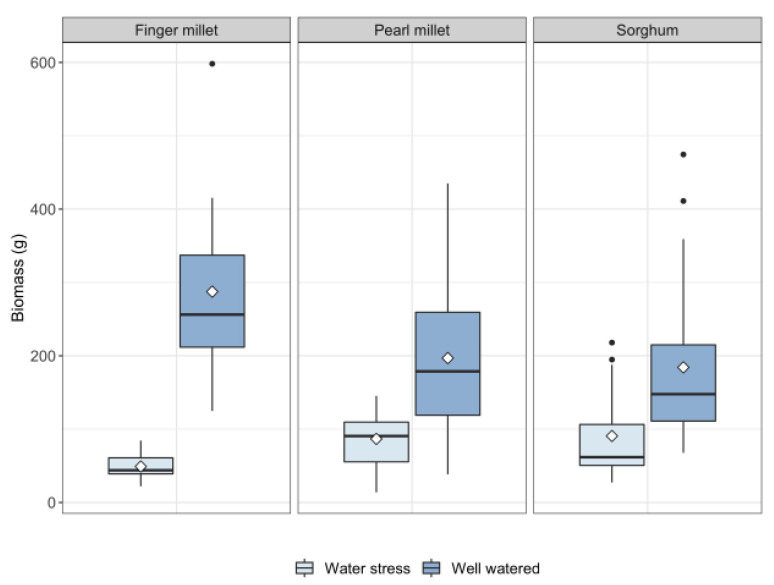
Boxplot of biomass production of finger millet, pearl millet and sorghum species. Biomass represents the sum of the weights of all the plant components: stem, leaves, panicles and tillers. Horizontal bar = median, white diamond = mean, black dots = outliers.

**Figure 2 plants-11-01019-f002:**
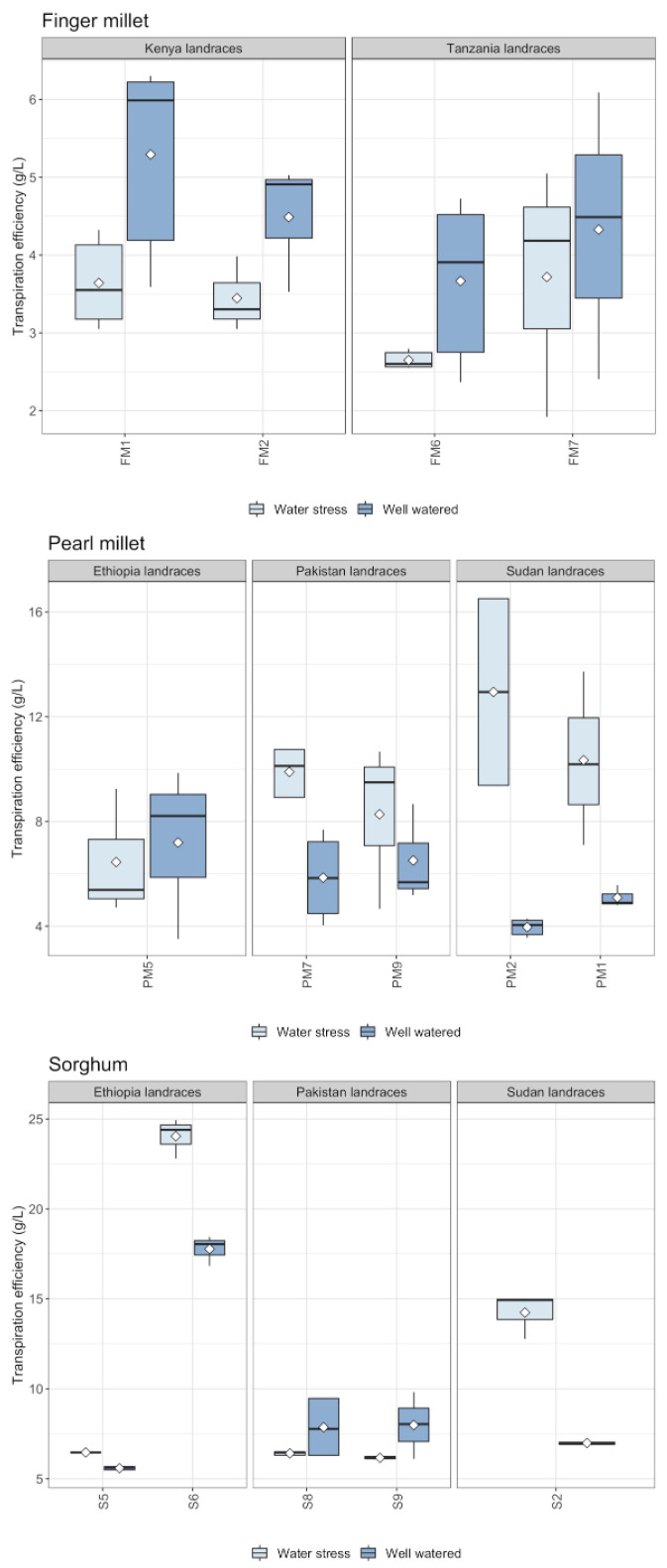
Boxplot of transpiration efficiency of finger millet, pearl millet and sorghum landraces by region of origin. Transpiration efficiency has been evaluated as the ratio between biomass produced/water transpired by the plant. Horizontal bar = median, white diamond = mean, black dots = outliers.

**Figure 3 plants-11-01019-f003:**
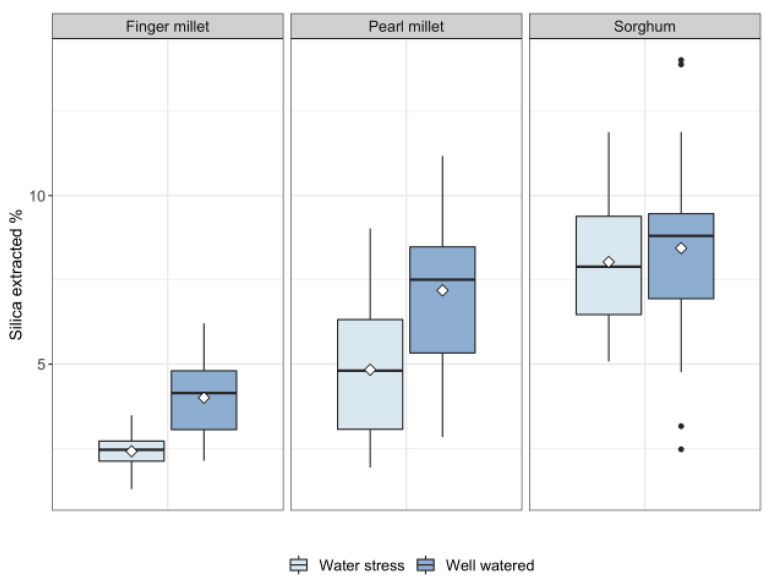
Boxplot of silica accumulation % values of finger millet, pearl millet and sorghum by water treatments. Horizontal bar = median, white diamond = mean, black dots = outliers.

**Figure 4 plants-11-01019-f004:**
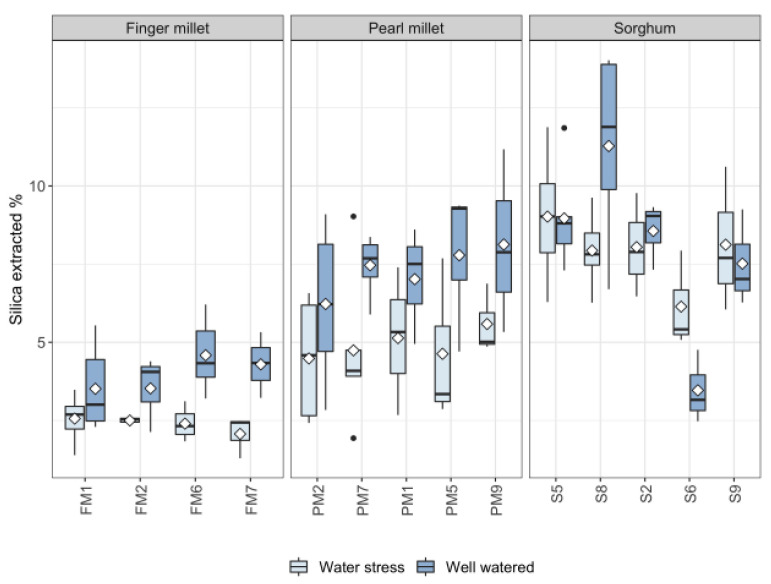
Boxplot of silica accumulation % values for the landraces of finger millet, pearl millet and sorghum by water treatments. Horizontal bar = median, white diamond = mean, black dots = outliers.

**Figure 5 plants-11-01019-f005:**
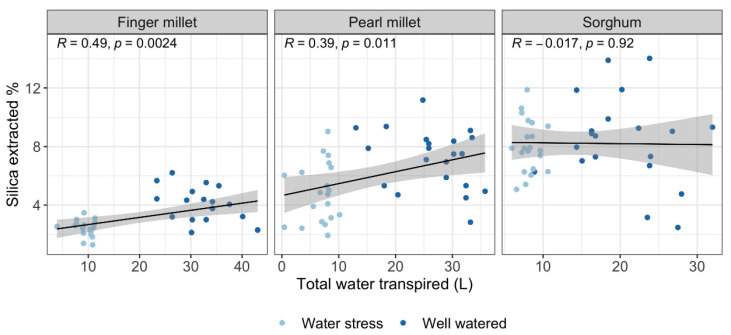
Linear regression tested on the 4 landraces of finger millet, 5 landraces of pearl millet and 5 landraces of sorghum. Total water transpired (L) is used as the independent variable, and % silica extracted from leaves is used as the dependent variable. Gray bands represent 95% confidence intervals.

**Figure 6 plants-11-01019-f006:**
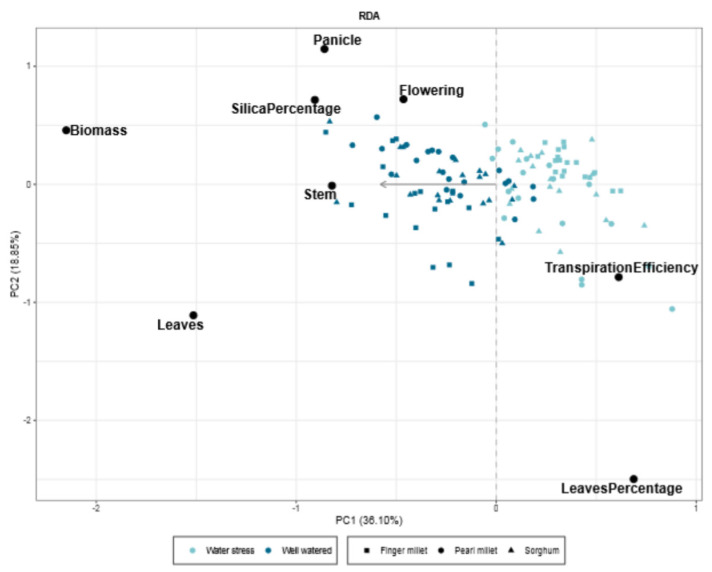
Triplot showing the results of the redundancy analysis. Gray arrow represents the explanatory variable (water treatment), black dots represent the variables included in the analysis (physiological parameters and biosilica accumulation), and blue spots indicate the samples which correspond to the single plant. Light blue = WS and blue = WW. Triangle = sorghum, circles = pearl millet, squares = finger millet.

**Table 1 plants-11-01019-t001:** Mean and standard deviation values for well-watered (WW) and water-stressed (WS) replications of the three crops.

	Total Water Transpired	Total Biomass	Leaf Biomass
**Sorghum**	WW: 20.38 ± 5.77 L WS: 8.19 ± 1.19 L	WW: 184.22 ± 112.63 g WS: 90.66 ± 60.70 g	WW: 24.02 ± 19.97 g WS: 14.75 ± 11.84 g
**Pearl millet**	WW: 26.78 ± 6.51 L WS: 6.72 ± 2.68 L	WW: 196.84 ± 100.57 g WS: 86.71 ± 38.81 g	WW: 10.76 ± 5.42 g WS: 8.03 ± 6.04 g
**Finger millet**	WW: 32.52 ± 5.88 L WS: 9.42 ± 1.87 L	WW: 287.42 ± 111.90 g WS: 49.10 ± 16.56 g	WW: 31.31 ± 11.38 g WS: 5.56 ± 1.71 g

**Table 2 plants-11-01019-t002:** Comparison of linear and multiple linear regression models to account for silica accumulation variability (% of silica extracted) in the dataset, including the three species of finger millet, pearl millet and sorghum.

Model Predictors	Adjusted R-Squared
Water treatment	0.050
Species	0.475
Genotype	0.540
Water treatment + Genotype	0.608
Water treatment × Genotype	0.635

**Table 3 plants-11-01019-t003:** Selected landraces from the ICRISAT genebank with a. the acronym used to identify them and b. their accession number. Climatic data are expressed as the annual mean. Precipitation and rainy days represent the total annual condition. Climatic indices are specific to the region of interest from which the samples come.

	Sudan	Ethiopia	Pakistan	Kenya	Tanzania
*Climatic data* Mean temperature Average sun hours Precipitation Rainy days Humidity	32.79 °C 10.9 h 70 mm 13 days 25.16%	27.63 °C 10.5 h 519 mm 60 days 37.16%	27.34 °C 10.7 h 152 mm 15 days 44.66%	30.03 °C 10.5 h 213 mm 25 days 47.91%	24.89 °C 10.3 h 602 mm 63 days 49.33%
Sorghum	S2: IS23075	S5: IS11061 S6: IS38025	S8: IS35215 S9: IS35216		
Pearl millet	PM1:IP13327 PM2: IP9859	PM5: IP2367	PM7: IP18019 PM9: IP18021		
Finger millet				FM1: IE2511 FM2: IE3476	FM6: IE4450 FM7: IE4456

## Data Availability

The data presented in this study are available in the article or its supplementary material.

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
