# Peer review of "Understanding the Relationship between Water Availability and Biosilica Accumulation in Selected C4 Crop Leaves: An Experimental Approach"

_plants, 2022, doi:10.3390/plants11081019_

Round 1
Reviewer 1 Report
Dear Authors, Dear Editor,
In general, the paper is well-written. Experimental approach, data evaluation, results and conclusion seem to be consistent and sound.
Some points remain unclear to me, but it shoud be easy to elucidate them:
329 "The experimental design included ten different landraces for each of the three species...", but Table 3. above shows only 14 ("selected") of them. What about the remaining 16 accessions?
334 "We set five replications for each landrace and treatment,..", whereas in 374 you wrote "For each landrace we analyzed samples coming from two plants of each of the three replications in both treatments". What was the reason for this reduction, as "All replications grew and produced leaves. (139)"?
I wonder, if the method that was applied for determination of silica in the palnts is actually state of the art (esp. in terms of selectivity). Anyhow, it should be described more exactly here (volumes, reaction time, separation technique (filtration/centrifugation?) itd. The link to the full protocol does not work.
Best regards
Author Response
We thank the reviewer for her/his comments and we revised the manuscript following thier suggestions:
329 "The experimental design included ten different landraces for each of the three species...", but Table 3. above shows only 14 ("selected") of them. What about the remaining 16 accessions?
The sentence line 329 has been modified:
329 “The experimental design included ten different landraces for each of the three species to obtain a sufficient sample size to observe the physiological parameters of the crops growing. Five genotypes of sorghum, five of pearl millet and four of finger millet were selected (Table 3) for the present study. The selection criteria was based on the physiological response to watering. The genotypes with highest diversity in physiological parameters were selected within each species in order to asses inter-genotype variations in biosilica accumulation. The remaining 16 landraces have been cropped and stored for future analysis.
334 "We set five replications for each landrace and treatment,..", whereas in 374 you wrote "For each landrace we analyzed samples coming from two plants of each of the three replications in both treatments". What was the reason for this reduction, as "All replications grew and produced leaves. (139)"?
The high number of samples (600 leaf samples in total), meant that we had to make a decision and reduce the sample size. 10 landraces and 5 replications each was the minimum sample size for reproducing field conditions in an experimental cultivation and to observe differences, if any, in the physiological response. We then selected the most physiological diverse landraces, with the idea of trying to avoid the loss of variability in biosilica accumulation. We reduced the number of analysed samples to three replicas to meet the statistical acceptance, based the selection on the number of samples tested in previous articles which highlighted differences in silica/phytoliths coming from experimentally growth crops (Jenikins et al., 2016 and Madella et al., 2009).
374 “For each landrace we selected and analyzed two plants of each of the three replications in both treatments. We selected three replications out of five to reduce the sample size while counting on more than 30 samples per species, as previously suggested to meet the statistical representativeness by Jenkins et al. (2016-2020).”
I wonder, if the method that was applied for determination of silica in the palnts is actually state of the art (esp. in terms of selectivity). Anyhow, it should be described more exactly here (volumes, reaction time, separation technique (filtration/centrifugation?) itd. The link to the full protocol does not work.
The procedure used to extract silica form modern leaf tissue, although based on common protocols in use within the phytolith community to extract biosilica from modern plant tissues, is experimental. The full protocol is currently under revision in journal that offered the possibility of publishing laboratory protocols. The manuscript has been modified adding the link to protocols.io where the complete steps have been described detailed, while the paper is still under process.
387 “The full protocol is available on protocols.io at this protocols.io

Reviewer 2 Report
The article “Understanding the relationship between water availability and biosilica accumulation in selected C4 crop leaves: an experimental approach” is generally well written. The text written by the authors meets the requirements of a scientific article. The experiment was conducted correctly, and the results are thoroughly described in this article. The discussion carried out is sufficient. I think the paper is suitable for publication; however, I would suggest rewriting the conclusion chapter. In its present form, the chapter does not highlight the results obtained by the authors and their novelty. I suggest including the most important conclusions from the experiment in this chapter in the form of a bulleted list.
Author Response
The conclusions have been modified, as suggested by the reviewer, to enhance the results achieved with the study. The changes made are:
406 “5. Conclusions
The results presented in this paper allows us to conclude that:
- Water availability plays a fundamental role in determining biosilica accumulation in finger millet and pearl millet, which seem to be passive accumulators where the transpiration-driven biosilica production prevails over the genetic mediated silica deposition. Therefore, we maintain that biosilica accumulation in finger millet and pearl millets is a good proxy for water availability.
- Based on the results obtained, different sorghum genotypes absorbed and accumulated silica differently. The relative high magnitude of variability in response to water treatment suggests that biosilica accumulation in sorghum is not a good proxy for plant water availability. Indeed, sorghum is seemingly characterized by a transporter-governed mechanism, which possibly determines a high variability among genotypes. In literature the topic is rather controversial. The results of this paper lead to new perspectives, highlighting that not all the sorghum genotypes respond equally to biosilica accumulation.
- Both environmental conditions and genetic variability play distinct roles in biosilica accumulation even within the same species.
Nevertheless, we want to highlight that different results have been published, especially in relation to archaeological studies, and this might derive form the experimental settings. Pot-based experiments, possibly conducted under light limitations within glasshouses, may be flawed because of the strong influence of light on transpiration stream and plant development. Therefore, we suggest that experimental cultivation using such a standardized methodology is now needed, also to respond to archaeological and palaeoenvironmental questions.”
